# A Methodological Framework for Assessing the Agroecological Performance of Farms in Portugal: Integrating TAPE and ACT Approaches

Inês Costa-Pereira [1,2,*] , Ana A. R. M. Aguiar [3] , Fernanda Delgado [4,5] and Cristina A. Costa [1,2]

1  IPV-ESA-Polytechnic Institute of Viseu, Agrarian School, Quinta da Alagoa Estrada de Nelas,
   3500-606 Viseu, Portugal; amarocosta@esav.ipv.pt
2  CERNAS-IPV-Centre for Natural Resources, Environment and Society, Polytechnic Institute of Viseu,
   Polytechnic Campus, 3504-510 Viseu, Portugal
3  GreenUPorto—Sustainable Agrifood Production Research Centre/Inov4Agro, DGAOT, Faculty of Sciences,
   Campus de Vairão, University of Porto, Rua da Agrária 747, 4485-646 Vairão, Portugal; aaguiar@fc.up.pt
4  IPCB-ESA-Polytechnic Institute of Castelo Branco, Agrarian School, Quinta da Senhora de Mércules,
   Apartado 119, 6001-909 Castelo Branco, Portugal; fdelgado@ipcb.pt
5  CERNAS-IPCB-Research Centre for Natural Resources, Environment and Society, Polytechnic Institute of
   Castelo Branco, 6001-909 Castelo Branco, Portugal
*  Correspondence: inespereira@esav.ipv.pt

**Abstract:** Agroecology integrates science, social movements, and agricultural practices, playing a central role in the sustainability of food systems. It addresses agroecosystems and food systems holistically; however, defining whether a farm is agroecological remains a challenge. This article proposes a methodology to measure farms' agroecological performance, adapted to the family farming context in Portugal. The aim of the developed methodology is to compare the agroecological performance of family farms (conventional and non-conventional), providing information about anchors for agroecological transition and supporting public policies. A literature review identified existing farm evaluation methodologies, with Tool for Agroecological Performance Evaluation (TAPE) and Agroecology Criteria Tool (ACT) scoring highest in an assessment process. Questions from both were integrated into a questionnaire for family farmers. This field work provided critical insights towards the methodologies: (1) territorial adaptability; (2) transition constraints' origin; and (3) use of non-academic language. The results were incorporated into the developed methodology, which combines the TAPE indicator matrix and Gliessman's five levels of food system change, the latter of which provides the framework for the ACT. This study made it possible to identify the most relevant aspects for characterizing family farmers/farms and the importance of how the evaluation criteria/indicators are ordered by element/theme, as it alters the values of each farm's agroecological performance.

**Keywords:** agroecology transition; family farms; territorial approaches; indicators for transition



## 1. Introduction

The impacts of conventional agriculture systems are recognized at the environmental level (biodiversity loss, water pollution, and scarcity of natural resources) [1–6], human health level (rates of malnutrition and hunger and intoxication of farmers and consumers) [7–10], and the level of rural territories (inadequate payments to farmers and rural abandonment) [11–13]. As an alternative path, agroecology has gained importance in regard to food systems' sustainability, and its approaches have come to prominence in technical, scientific, and political discourses [14–16]. However, agroecology must be supported by public policy that is co-constructed and adapted to each territory as a tool for the transition to more sustainable food systems [17,18].

Agroecology merges sustainable agricultural practices, social movements, and science; and it is more effective when these three dimensions converge [19,20]. The concept of agroecology is dynamic, holistic, and should be applied to the food system, from production to consumption, by integrating action and change aimed towards economic, ecological, and social sustainability. As a social movement, agroecology focuses on creating local food systems that support rural communities' sustainability by promoting safe food production practices, short food supply chains, and food sovereignty. As a set of agricultural practices, agroecology aims to improve agricultural systems based on natural processes, enhancing interactions and strengthening beneficial synergies between the components of the agroecosystem, using ecological processes and ecosystem services as tools, and reducing dependence on off-farm inputs. As a science, agroecology integrates research, education, action, and change [21,22].

Most agroecological farmers are smallholders and/or family farmers that use traditional and environmentally friendly farming techniques/technologies, such as crop selection and rotation, intercropping, agroforestry, or mulching. They use inputs produced on the farm, and they do not use synthetic fertilizers or pesticides favoring organic fertilization and biological pest control, among other practices. Frequently, agroecological farmers use innovative practices as reduced tillage, drip irrigation, or direct seeding [22,23]. These farmers frequently connect through social networks for the co-creation and exchange of agroecological knowledge and are involved in local food systems [16,22].

For the purpose of this study, we analyzed family farms, as they represent, in Portugal, more than 90% of the country's single farmers [24]. FAO & IFAD [25] defined family farms as those managed and operated by a family that lives ond the farm and provide the majority of the farm labor. Moreover, family farming is recognized as going beyond the productive or economic perspective of modern and intensive agriculture because it preserves and transmits local knowledge, protects the natural resources, and ensures food security for the families [26].

Recent studies, carried out in the north and center of Portugal, reveal that family farms maintain a majority of sustainable practices. However, to sustain the agroecological transition, it is necessary to identify and overcome compromising practices that are detrimental to public and environmental health—e.g., the use of synthetic pesticides and fertilizers [16,27–30]. To support this transition, it is necessary to assess the family farms' agroecological performance, identifying factors that: (1) can anchor that transition; (2) raise the farmers' willingness to start the transition process; and (3) advocate for public policies that promote the agroecological transition on Portuguese farms.

Although there are several methodologies to evaluate the farms' agroecological performance or the farms' sustainability, their limitations have been identified. The research done by authors such as Widget [31], de Olde [32], Bonisoli [33], or Nicholls [34] state that these methodologies are usually designed to be applied only to a geographical region and/or are based on indicators unable to capture the multifunctionality of the agroecosystem (e.g., economic function or food-production indicators).

Thus, to support the design of public policies that promote the family farms' agroecological transition in Portugal, a defined methodology is needed to analyze agricultural systems with regard to their territorial context, their role in rural territories, and their pivotal connection with urban territories. It is also necessary to emphasize that, although addressed at a local/national level, the important role of family farming and its lack of support is a global problem that needs to be addressed collectively to give it visibility and strength. This study presents a methodology designed to be used in farms (also) in European contexts, one focused on assessing their agroecological performance by providing information on the anchors for the agroecological transition—the baseline information for the development of co-constructed guidelines for adequate public policies for the agroecological transition.

## 2. Materials and Methods

To develop the farms agroecological-performance methodology, a systematic literature review was conducted between September and December 2021 using the web platforms and databases Scopus, Web of Science, and Google Scholar. Several combinations of keywords were used, including "agroecology", "agroecological", "sustainable", "transition", "assessment", "tool", and "method", connected with the boolean operators "AND" and "OR". This thorough search turned up 12 different combinations.

The main objective of the literature review was to identify the most cited and recognized methodologies for the evaluation of the farms' agroecological or sustainability performance. A total of 123 records were identified from the referred databases and 27 were selected, based on inclusion and exclusion criteria. The inclusion criteria were: English-language, original articles, published since 2010, focused on methodologies for the evaluation of the agroecological or sustainability performance of farms, and the evaluation of those methodologies. The exclusion criteria were: grey literature on the topics, original articles off topic, and full-texts not available. Of the 27 scientific articles selected, 10 methodologies were compiled: Agroecology Criteria Tool (ACT) and Farm Level Agroecology Criteria Tool (F-ACT), the Indicateurs de Durabilité des Exploitations Agricoles (IDEA), Life Cycle Assessment in Agriculture (LCA-A), Marco para la Evaluacion de los Sistemas de Manejo de Recursos Naturales (MESMIS), Multi-Level Perspective (MLP), Response-Inducing Sustainability Evaluation (RISE), Rural Household Multi-Indicator Survey (RHoMIS), Tool for Agroecology Performance Evaluation (TAPE), and Sustainability Assessment of Food and Agriculture Systems (SAFA).

To analyze the 10 selected methodologies, a procedure for their evaluation was developed based on the studies carried out by Binder [35], de Olde [32], Talukder & Blay-Palmer [36], and Bonisoli [33], related to their normative dimension (focused on the methodology evaluation) and procedural dimension (focused on how the assessment was carried out). The criteria selected were those that best responded to the objective and purpose of the methodology to be developed. For the normative dimension, the goal-setting approach; for the procedural dimension, the methodology simplicity, time requirement, assessment purpose, indicators and attributes, target group, stakeholders' participation, geographical application, and assessment level; as shown in Table 1.

The evaluation of the 10 selected methodologies, using the normative and procedural dimensions, consisted of assigning a value of 0 (zero) when the criteria was absent and 1 (one) when it was present. The methodologies with the highest scores were those with the presence of most of the criteria selected. The results of the methodologies' evaluation are presented in Table 2, were each column sums the results obtained from each criterion (e.g., for the criteria "methodology simplicity", ACT scored 1 in "graphic visualization of the results", "free access to software", "easy access to software", and "free access to tutorials" resulting in a total score of 4). The methodologies with the highest scores (TAPE and ACT) were the ones selected for the next step.

**Table 1.** Methodology analyses framework with the dimensions, criteria valuations, and justifications.

| Dimension | Criteria Valuation: Yes (1)/No (0) | Criteria Justification | References |
|---|---|---|---|
| Normative Dimension: scope of the methodology evaluation | Goal setting approach: bottom-up (stakeholder); top-down (theory) | The definition of the methodology goals provides the basis of and operationalizes the farms assessment. It is important to understand whether the methodology goals and criteria are: (a) predefined and theoretical, derived from a definition of agroecology/sustainability (top-down approach); (b) defined by stakeholders in a participatory process (bottom-up approach). The combination of top-down and bottom-up approaches was assumed to be a transdisciplinary approach | [33,35] |

**Table 1.** *Cont.*

| Dimension | Criteria Valuation: Yes (1)/No (0) | Criteria Justification | References |
|---|---|---|---|
| Procedural Dimension: how the assessment was carried out | Methodology simplicity: graphic visualization; free access; easy to access; free access to the tutorials | The methodology simplicity, or its user friendliness, was assessed based on: (a) graphic visualization of the results that simplifies the end users' (farmers and policy makers) understanding; (b) access to free methodology software that allows a fast and automatic calculation of the results and improves the communication between users; (c) in some cases, the methodologies are available for free, but it is necessary that some procedures are accessible; (d) free access to tutorials to help the simplify the methodology's application. | [36] |
| | Time requirement: more than 5 h; between 3 and 5 h; less than 3 h | The time required for the methodology application is directly connected to the effort needed to get to know and apply the methodology and discuss the results. If a methodology is too time consuming, that it can be an inhibiting factor for its success. | [32] |
| | Assessment purpose: self-assessment and guidance; assessment and guidance; multiple purpose | Refers to the aim for which the tool was created, which can be for: (1) self-assessment and guidance: intended to be used by farmers themselves; (2) assessment and guidance: developed with the purpose of helping farmers assess and pursue sustainability/agroecology in their farms; (3) multiple purpose: designed for multiple stakeholders or developed with multiple purposes, without a specific sustainability/agroecology target. | [35] |
| | Indicators' attributes: predetermined; specific criteria; selection presented; interaction: | The availability of information about the indicators, including whether they are predefined, why they were selected, and whether there is an interaction between indicators, allows for assessing the robustness of the methodology and guarantees its replicability. | [33,35] |
| | Indicators' reference values: target values; threshold values; relative values | Reference values might be defined to assess the level of sustainability for a set of indicators. They can be absolute (target value or threshold) or relative: (1) target values: identify a desirable condition; (2) threshold values: define a minimum and maximum acceptable level; (3) relative values: compare indicators with an initial value, regional or sample average, or desirable trend. | [32,36] |
| | Target group: farmers; policy makers; researchers; educators | Target groups can range from farmers to educators. It is important to know the target group(s) of the methodology, and to understand if key elements for the analysis may be missing. | [35] |
| | Stakeholders' level of participation: whole process; partial; none | There are different degrees of stakeholder involvement in the methodology's application: (1) stakeholders can play a central role in the development, application, and interpretation of the indicators; (2) stakeholders can be consulted in specific parts of the methodology (development, application, or interpretation); (3) or their participation is not taken into account. | [35] |
| | Application in: Global North; Global South | Some methodologies were developed for specific socio-geographic contexts and are not suitable for replication in different socio-geographic contexts (e.g., developed and underdeveloped countries) | [32] |

Based on the content of the selected methodologies (TAPE and ACT), a set of questions was integrated in the questionnaire applied to family farmers. This questionnaire was organized to respond the TAPE Grid of Indicators (Step1) and Evaluation Criteria (Step 2)

and to the ACT questions, aiming to verify the suitability of these methodologies to assess Portuguese family farmers' agroecological performance.

TAPE was created by the Food and Agriculture Organization (FAO) to evaluate different production systems, including forestry, aquaculture, fisheries, and agricultural production, adaptable to different cultural and context circumstances. Four steps make up the TAPE process. STEP 0: gather relevant data on the area of study, important for step 3. STEP 1: evaluation, through the application of a questionnaire to farmers and a grid of 35 indicators, of the FAO's 10 elements of agroecology: recycling, efficiency, diversity, resilience, synergies, co-creation and sharing of knowledge, cultural and food traditions, human and social values, circular and solidarity economy, and responsible governance. The grid of indicators assessment is based on descriptive scales for evaluation ranging from 0 (indicative of lower agroecological performance) to 4 (indicative of higher agroecological performance). STEP 2: analysis of the 10 Evaluation Criteria: agrobiodiversity, soil health, production, income, added value, exposure to pesticides, dietary diversity, women's empowerment, youth job opportunities, and land tenure. These criteria are categorized into three levels: desirable, acceptable, and undesirable. STEP 3: participatory analysis of the results together with the farmers.

The ACT was developed by the Foundation for Ecological Development (Biovision). It is rooted in Gliessman's [37] categorization of the five levels representing the transition of food systems. These levels involve (1) enhancing the efficiency of conventional and industrial practices, (2) substituting conventional or industrial inputs with more sustainable alternatives, (3) redesigning entire agroecosystems, (4) reestablishing connections between growers and consumers through the development of alternative food networks, and ultimately, (5) restructuring the global food system to ensure sustainability and equity for all. These levels are aligned with eleven elements: regulation and balance, plus FAO's 10 elements of agroecology. The transition criteria defined (62 in total), can be answered with a 0 (if absent) or a 1 (if present). The resulting evaluations can be presented individually or collectively for a range of farms.

The questionnaire was applied, in a face-to-face interview, to eight family farmers between April and August 2022 in the Centre (Fundão, Mangualde, Viseu, and Vouzela municipalities) and South (Arraiolos and Évora municipalities) of Portugal (Figure 1). The contact with the family farmers was done through partnerships with local development organizations and farmers' associations.

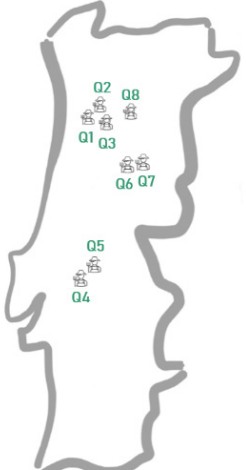

**Figure 1.** Localization of the farms (Q1–Q8) where the selected methodologies were applied.

## 3. Results

In the evaluation of the normative and procedural dimensions of the 10 selected methodologies for the evaluation of the farms' agroecological or sustainability performance, the highest score was 18 points, achieved by both TAPE and ACT (Table 2).

**Table 2.** Analysis of the 10 methodologies for farms' sustainability/agroecology assessment. Valuation criteria were scored as Yes (1)/No (0). Each cell presents the result obtained on each criterion (e.g., on the criterion "methodology simplicity", ACT scored 1 in "graphic visualization of the results", "free access to software", "easy access to software", and "free access to tutorials", making its score 4 in total).

| Methodology | Normative Dimension | Procedural Dimension | | | | | | | | | Total | References |
| | Goal Setting Approach: Bottom-Up (Stakeholder) OR Top-Down (Theory) | Methodology Simplicity: Graphic Visualization of the Results AND/OR Free Access to the Software AND/OR Easy to Access the Software AND/OR Free Access to the Tutorials | Time Requirement: More than 5 h OR between 3 and 5 h OR Less than 3 h | Assessment Purpose: Self-Assessment and Guidance OR Assessment and Guidance OR Multiple Purpose | Indicators' Reference Values: Target Values AND/OR Threshold Values AND/OR Relative Values | Indicators' Attributes: Predetermined Indicators AND/OR Specific Criteria for the Indicators' Selection AND/OR Indicators' Interaction | End Users: Farmers AND/OR Policy Makers AND/OR Researchers AND/OR Educators | Stakeholders' Level of Participation: Whole Process OR Partial OR None | Applied on: Global North AND/OR Global South | Assessment Level (Scale): Food System AND/OR Farm System OR Specific Productions | | |
|---|---|---|---|---|---|---|---|---|---|---|---|---|
| ACT (Agroecology Criteria Tool) | 1 | 4 | 1 | 1 | 1 | 3 | 2 | 1 | 2 | 2 | 18 | [38–40] |
| Farm Level Agroecology Criteria Tool (F-ACT) | 1 | 4 | 1 | 1 | 1 | 3 | 1 | 1 | 1 | 1 | 15 | [41,42] |
| IDEA (The Indicateurs de Durabilité des Exploitations Agricoles) | 1 | 2 | 1 | 2 | 1 | 1 | 4 | 1 | 2 | 1 | 16 | [32,33,35,43–46] |
| Life Cycle Assessment—Agricultural | 1 | 0 | 1 | 1 | 1 | 2 | 1 | 1 | 2 | 1 | 11 | [47,48] |
| MLP (Multi-Level Perspective) on agro-food sustainability transitions | 1 | 0 | 1 | 1 | 1 | 0 | 1 | 1 | 2 | 2 | 10 | [49–54] |
| MESMIS (Marco para la evaluacion de los sistemas de manejo de recursos naturales) | 1 | 1 | 1 | 1 | 1 | 1 | 3 | 1 | 1 | 2 | 13 | [33,36,55,56] |
| RISE (Response-Inducing Sustainability Evaluation) | 1 | 3 | 1 | 1 | 1 | 2 | 3 | 1 | 2 | 1 | 16 | [32,33,44,46,57] |
| RHoMIS (Rural Household Multi-Indicator Survey) | 1 | 3 | 1 | 1 | 1 | 2 | 2 | 1 | 1 | 1 | 14 | [58–62] |

**Table 2.** *Cont.*

| Methodology | Normative Dimension | Procedural Dimension | | | | | | | | | | |
|---|---|---|---|---|---|---|---|---|---|---|---|---|
| | Goal Setting Approach: Bottom-Up (Stakeholder) OR Top-Down (Theory) | Methodology Simplicity: Graphic Visualization of the Results AND/OR Free Access to the Software AND/OR Easy to Access the Software AND/OR Free Access to the Tutorials | Time Requirement: More than 5 h OR between 3 and 5 h OR Less than 3 h | Assessment Purpose: Self-Assessment and Guidance OR Assessment and Guidance OR Multiple Purpose | Indicators' Reference Values: Target Values AND/OR Threshold Values AND/OR Relative Values | Indicators' Attributes: Predetermined Indicators AND/OR Specific Criteria for the Indicators' Selection Presented AND/OR Indicators' Interaction | End Users: Farmers AND/OR Policy Makers AND/OR Researchers AND/OR Educators | Stakeholders' Level of Participation: Whole Process OR Partial OR None | Applied on: Global North AND/OR Global South | Assessment Level (Scale): Food System AND/OR Farm System OR Specific Productions | Total | References |
| TAPE (Tool for Agroecology Performance Evaluation) | 1 | 2 | 1 | 1 | 2 | 3 | 3 | 1 | 2 | 2 | 18 | [38,63–65] |
| SAFA (Sustainability Assessment of Food and Agriculture Systems) | 1 | 2 | 1 | 1 | 1 | 2 | 3 | 0 | 2 | 2 | 15 | [32,33,35,44,66] |

The questionnaire, prepared with TAPE and ACT questions and criteria, was applied to eight farmers (six men and two women) ranging from 32 to 85 years old. Their educational backgrounds were diverse, ranging from fourth-grade educations to advanced degrees in subjects like philosophy, history, or the arts. Only one farmer had a degree in agronomy (Q5). The farmers had a variety of employment situations: four were retired, two were self-employed with employees, one was self-employed without employees (Q8), and one was an employee in other company (Q2). All farms were polycultural: seven farms included both animals and vegetables, and one included vegetables and olive trees (Q6). One farm does regenerative agriculture (Q8). The size of the farms ranged from 1 to 180 hectares, with the number of parcels ranging from 1 to 30 (Table 3).

**Table 3.** Farms (Q1 to Q8) and farmers' characterization.

| | Farm | | | | Farmer | | | |
|---|---|---|---|---|---|---|---|---|
| | Localization (Municipality) | Type of Products | Type of Agriculture | Farm Size | Gender | Age | Education Level | Job Situation |
| Q1 | Vouzela (Centre of Portugal) | milk production/subsistence vegetable garden | conventional | 1 ha/1 parcel | man | 67 | basic school | self-employed with employees |
| Q2 | Vouzela (Centre of Portugal) | livestock/subsistence vegetable garden | conventional | 6 ha/ 30 parcels | man | 54 | primary school | employed in another company |
| Q3 | Viseu (Centre of Portugal) | livestock/subsistence vegetable garden | conventional | 6 ha/30 parcels | human | 72 | primary school | retired |
| Q4 | Arraiolos (South of Portugal) | vegetables/sheep | conventional | 3 ha/2 parcels | man | 85 | primary school | retired |
| Q5 | Évora (South of Portugal) | vegetables/sheep | conventional | 180 ha/ 8 parcels | man | 51 | degree in agronomic engineering | self-employed with employees |
| Q6 | Fundão (Centre of Portugal) | vegetables/olives | conventional | 3 ha/1 parcel | man | 76 | degree in philosophy | retired |
| Q7 | Fundão (Centre of Portugal) | vegetables/livestock | conventional | 2 ha/ 25 parcels | man | 66 | degree in history | retired |
| Q8 | Mangualde (Centre of Portugal) | vegetables/livestock | regenerative agriculture | 10 ha/ 1 parcel | human | 32 | degree in arts | self-employed with no employees |

Analyzing the results obtained on the eight farms by applying the TAPE Grid of Indicators (Table 4), Farm Q8 stands out in terms of agroecological performance, boasting an average rating of 2.8 (of a possible total of 4.0 values). Following Q8 closely is Farm Q7 (2.4 points). Farm Q8's distinction lies in its proficiency across a multitude of indicators. It obtained the highest score of 4 points on seven indicators, including 'Management of Soil Fertility' and 'Management of Pest and Diseases', and the second-highest score of 3 points on 16 indicators. Farm Q7 obtained a maximum score on two indicators, 'Appropriate Diet and Nutritional Awareness' and 'Local or Traditional Identity and Awareness', and secured a score of 3 on 15 indicators.

On the other hand, Farm Q6 presented the lowest recorded value (1.6), with a score of either 0 or 1 on eight indicators.

The assessment of agroecological performance across the surveyed farms reveals noteworthy trends. The highest average score of 3.0 was observed in two key areas: in 'Crops indicator from Diversity Element', indicating a consistent adoption of polycultural practices across all farms; and in 'Local or Traditional Identity' and 'Awareness from the Culture and Food Tradition Element'. The latter, together with the indicator 'Appropriate Diet and Nutrition Awareness', garners the second-highest average score of 2.9, emphasize the significant influence of local identity and cultural context on dietary choices. Within the 'Circular and Solidarity Economy Element', the indicator measuring 'Networks of Producers, Relationship with Consumers, and Presence of Intermediaries' also attains an average score of 2.9. Indeed, the existence of communication channels between producers and consumers, and the participation of agricultural or local development associations, has facilitated the contact with the family farmers interviewed.

**Table 4.** Results of the TAPE grid of indicators' application in the eight farms in the center and south of Portugal, in 2022. The value scored by each farm varies from 0 (less agroecological) to 4 (more agroecological).

| FAO 10 Elements of Agroecology | Indicators | Q1 | Q2 | Q3 | Q4 | Q5 | Q6 | Q7 | Q8 | Average Value per Indicator |
|---|---|---|---|---|---|---|---|---|---|---|
| Diversity | Crops | 3 | 2 | 3 | 3 | 3 | 3 | 3 | 4 | 3.0 |
| | Animals (including fish and insects) | 1 | 1 | 3 | 2 | 3 | 0 | 2 | 3 | 1.9 |
| | Trees (and other perennials) | 1 | 2 | 2 | 3 | 2 | 2 | 3 | 4 | 2.4 |
| | Diversity of activities, products, and services | 1 | 0 | 2 | 1 | 2 | 1 | 2 | 2 | 1.4 |
| Synergies | Crop–livestock–aquaculture integration | 1 | 3 | 2 | 2 | 2 | 0 | 2 | 2 | 1.8 |
| | Soil–plants system management | 4 | 3 | 2 | 3 | 2 | 0 | 2 | 3 | 2.4 |
| | Integration with threes | 1 | 1 | 2 | 2 | 2 | 2 | 2 | 3 | 1.9 |
| | Connectivity between elements of the agroecosystem and the landscape | 1 | 3 | 3 | 3 | 2 | 1 | 2 | 3 | 2.3 |
| Efficiency | Use of external inputs | 2 | 3 | 1 | 2 | 2 | 1 | 2 | 3 | 2.0 |
| | Management of soil fertility | 4 | 4 | 1 | 2 | 2 | 1 | 3 | 4 | 2.6 |
| | Management of pests and diseases | 0 | 4 | 1 | 1 | 0 | 0 | 2 | 4 | 1.5 |
| | Productivity and household needs | 1 | 3 | 2 | 2 | 3 | 2 | 2 | 2 | 2.1 |
| Recycling | Recycling of biomass and nutrients | 4 | 3 | 2 | 2 | 3 | 0 | 1 | 3 | 2.3 |
| | Water saving | 0 | 2 | 1 | 1 | 2 | 2 | 3 | 3 | 1.8 |
| | Management of seeds and breeds | 1 | 3 | 3 | 3 | 1 | 1 | 1 | 3 | 2.0 |
| | Renewable energy and production | 0 | 0 | 1 | 0 | 1 | 0 | 1 | 2 | 0.6 |
| Resilience | Stability of income/production and capacity to recover from perturbations | 2 | 3 | 1 | 3 | 2 | 2 | 3 | 4 | 2.5 |
| | Mechanisms for reducing vulnerability | 4 | 3 | 0 | 0 | 3 | 3 | 3 | 2 | 2.3 |
| | Environmental resilience and capacity to adapt to climate change | 0 | 1 | 1 | 2 | 2 | 2 | 2 | 2 | 1.5 |
| Culture and Food Tradition | Appropriate diet and nutrition awareness | 3 | 3 | 2 | 3 | 3 | 3 | 4 | 2 | 2.9 |
| | Local or traditional identity and awareness | 3 | 3 | 2 | 3 | 3 | 3 | 4 | 3 | 3.0 |
| | Use of local varieties/breeds and traditional knowledge for food preparation | 2 | 2 | 3 | 3 | 2 | 3 | 3 | 2 | 2.5 |
| Co-creation and Sharing of Knowledge | Platforms for the horizontal creation and transfer of knowledge and good practices | 0 | 0 | 1 | 0 | 2 | 0 | 0 | 3 | 0.8 |
| | Access to agroecological knowledge and interest of producers in agroecology | 0 | 0 | 0 | 0 | 1 | 1 | 2 | 3 | 0.9 |
| | Participation of producers in networks and grassroots organizations | 2 | 3 | 2 | 3 | 3 | 3 | 3 | 2 | 2.6 |
| Human and Social Values | Women's empowerment | 3 | 1 | 2 | 2 | 2 | 3 | 2 | 3 | 2.3 |
| | Labor (productive conditions, social inequalities) | 2 | 1 | 0 | 2 | 3 | 2 | 2 | 2 | 1.8 |
| | Youth empowerment and emigration | 1 | 0 | 0 | 0 | 2 | 1 | 1 | 4 | 1.8 |
| | Animal welfare [if applicable] | 3 | 3 | 3 | 3 | 3 | NA | 3 | 3 | 2.1 |

**Table 4.** *Cont.*

| FAO 10 Elements of Agroecology | Indicators | Q1 | Q2 | Q3 | Q4 | Q5 | Q6 | Q7 | Q8 | Average Value per Indicator |
|---|---|---|---|---|---|---|---|---|---|---|
| Circular and Solidarity Economy | Products and services marketed locally | 4 | 0 | 1 | 4 | 3 | 0 | 3 | 3 | 2.3 |
| | Networks of producers, relationships with consumers, and presence of intermediaries | 3 | 0 | 2 | 4 | 4 | 3 | 3 | 4 | 2.9 |
| | Local food system | 3 | 2 | 1 | 2 | 2 | 1 | 1 | 2 | 1.8 |
| Responsible Governance | Producers' empowerment | 1 | 1 | 1 | 2 | 3 | 3 | 3 | 1 | 1.9 |
| | Producers' organizations and associations | 2 | 1 | 3 | 1 | 3 | 3 | 3 | 3 | 2.4 |
| | Participation of producers in governance of land and natural resources | 0 | 2 | 3 | 0 | 0 | 3 | 3 | 0 | 1.4 |
| | Average Value Per Farm | 1.8 | 1.9 | 1.7 | 2.0 | 2.2 | 1.6 | 2.3 | 2.7 | |

Conversely, the lowest average score of 0.6 pertains to the 'Renewable Energy and Production' indicator within the 'Recycling' element. This suggests limited investment by the surveyed farmers in renewable energy production. Additionally, within the 'Co-creation and Sharing of Knowledge' element, two indicators, namely, 'Platforms for Horizontal Creation and Transfer of Knowledge and Good Practices' (scoring 0.8), and 'Access to Agroecological Knowledge and Producers' Interest in Agroecology' (scoring 0.9), had the lowest average scores. Despite their participation in producer networks, these farmers do not seem to perceive these platforms as conducive to knowledge exchange or co-creation. This is in line with the general perception that traditional cooperatives have failed, due to structural problems resulting from factors such as the atomization of farms, reduced innovation, low productivity, the advanced age of farmers, low levels of education, and aversion to risk [67].

Moreover, while certain practices closely align with agroecology, such as the adoption of polyculture and the promotion of local culture and traditions, the majority of the interviewed farmers remain unfamiliar with the conceptual framework of agroecology. Similar results were obtained in other Portuguese studies that evaluated the adoption of organic or regenerative practices [28,68,69]. This analysis underscores the fledgling status of agroecology in Portugal, particularly in light of the advanced age demographic of farmers (average age of 64), and their low education level (46.3% only completed primary school and 53.0% have only practical agricultural training) [70]. These demographic characteristics should be taken into consideration when formulating policies, programs, and initiatives, emphasizing the need for targeted efforts to disseminate and promote agroecological principles within the agricultural community.

The values obtained by Farm Q8 reveal important disparities. The lowest values obtained (0 and 1) are related to the lack of governance processes in the area where the farm is located (Mangualde municipality, in the Centre of Portugal). These results reveal the difficulty that farmers, especially family farmers, face in participating in governance structures even when there is still some support from local associations. The highest values (a value of 4) obtained by Farm Q8 are distributed among the elements 'Diversity' (the variety of crops and tree species planted is significant), 'Efficiency' (the management of soil fertility and of pests and diseases, using biological practices), 'Resilience' (income and production are stable and therefore able to increase annually, and there is the ability to recover fully and quickly after shocks/disturbances), 'Human and Social Values' (farming family of an young couple who see their future in farming and are keen to continue and improve their activity), and 'Circular and Solidarity Economy' (the farmers are part of a network of producers and have their own network of consumers—a Community Supporting Agriculture—CSA—to whom they deliver weekly baskets).

The analysis of the indicators of the FAO's 10 Elements of Agroecology (Step 1) reveals the intricate array of factors influencing the agroecological performance of a farm. Summarizing the key findings for each of the 10 elements:

(a) Diversity: As the farms are polycultural, they exhibit high degrees of diversity. However, the reduced diversity of practices, products, and services—in particular, that of Farms Q1, Q2, and Q6—negatively impacts their overall performance in this element.

(b) Synergies: The results obtained on the eight Farms are variable, meaning that there is some connectivity/interaction between crops/animals/landscape, although this is not consistent on the majority of the farms. The synergies that exist between the different components of production (on and off the farm) are very important and can be the lever for the agroecological transition of a farm.

(c) Efficiency: The use of synthetic pesticides and fertilizers, and the dependence on outside inputs, results in low values for the 'Management of Pests and Diseases' indicator for Farms Q1, Q5, and Q6. On all the farms surveyed, the valorization of residues may be improved.

(d) Recycling: Most farms have minimal energy production; instead, they often acquire energy from the market. Only two farms (Q7 and Q8) are identified as using water-saving techniques, information that underlines the importance of tackling resource management in the context of agroecological transition.

(e) Resilience: Most agroecosystems have a good capacity to adapt to climate irregularities, except Farm Q3, which faces challenges in income stability, vulnerability reduction mechanisms, and environmental resilience.

(f) Culture and Food Tradition: The majority of farmers demonstrate a strong awareness of local traditions and heritage, and a commitment to preserving them through traditional recipes and participation in local fairs and festivities.

(g) Co-creation and Sharing of Knowledge: While farmers are involved in networks and grassroots organizations, they generally have limited involvement in co-creation and knowledge-sharing initiatives. Awareness of agroecological principles and concepts is low for the majority of the famers interviewed. This factor can be related to the farmers' age and education levels.

(h) Human and Social Values: It is difficult to get young people interested in agriculture, and they are rarely present on most farms. Only Farm Q6 has employed a young woman with an interest in working in agriculture, specifically in animal production.

(i) Circular and Solidarity Economy: Six farms participate in networks of producers and consumers, selling locally and engaging with consumers. Only one farmer (Q8) is part of a CSA.

(j) Responsible Governance: Farmers rarely participate in governance processes, which points to a lack of local mechanisms for participation.

This detailed analysis provides a comprehensive understanding of the agroecological performance of and challenges faced by the surveyed family farms. It highlights areas where improvements can be made to enhance agroecological practices and sustainability. It also underscores the importance of addressing issues such as knowledge sharing, resource management, and youth engagement in agriculture to promote agroecological transition.

Concerning the application of the TAPE Step 2 (Evaluation of the 10 Criteria), Farms Q2 and Q8 had the most favorable outcomes, or "green lights" in 'Land Tenure', 'Income', 'Added Value', and 'Dietary Diversity'; Farm Q2 also achieved the best outcomes for 'Productivity' and 'Soil Health', and Farm Q8 for 'Pesticide Exposure' and 'Agricultural biodiversity'. In contrast, Farm Q5 showed a higher percentage of adverse indicators, or "red lights", in 'Income', 'Pesticide Exposure', 'Women's Empowerment', 'Youth Employment', and 'Agricultural biodiversity' (Table 5).

**Table 5.** Results of the application of the 10 Evaluation Criteria of TAPE (STEP 2) on the eight farms, in the center and south of Portugal, in 2022. In the table are the farms (Q1 to Q8). The value scored by each farm varies between 3 levels—desirable (green-G), acceptable (yellow-Y), and undesirable (red-R).

| TAPE Traffic Light Approach | Q1 | Q2 | Q3 | Q4 | Q5 | Q6 | Q7 | Q8 |
|---|---|---|---|---|---|---|---|---|
| Secure land tenure | G | G | G | G | G | G | G | G |
| Productivity | Y | G | R | G | Y | Y | G | R |
| Income | G | G | G | Y | R | Y | Y | G |
| Value added | G | G | R | G | G | G | G | G |
| Exposure to pesticides | R | R | G | R | R | R | Y | G |
| Dietary diversity | G | G | G | Y | G | G | G | G |
| Women's empowerment | R | R | Y | R | R | R | R | Y |
| Youth employment opportunity | R | R | R | R | R | R | R | R |
| Agricultural biodiversity | R | R | R | R | R | Y | G | G |
| Soil health | G | G | G | Y | G | Y | Y | Y |

Summarizing the key findings for each of the 10 criteria:

(a) Land Tenure: No farms experience insecurity regarding land tenure, but there is a general concern about who will take over the land in the future.

(b) Productivity: Rather than the size or diversity of the farm, productivity varies. Some farms face challenges in registering their productivity due to the complexity of their activities.

(c) Income: For most farms, income has either remained stable or improved compared to previous years. Only one large-scale Farm (Q5) experienced a loss in income, most likely as a result of difficulties in the livestock sector.

(d) Added Value: Only one Farm (Q3) received a red light, indicating challenges in 'Adding Value', which can be explained by the farmer's health problems and the cost of maintaining the animals.

(e) Exposure to Insecticides: Only Farms Q3 and Q8 received a green light, as they use natural pesticides or implement biological crop protection practices.

(f) Dietary Diversity: Farm Q4 received a yellow light due to the farmer's limited diet, likely influenced by their advanced age.

(g) Women's Empowerment: Farms (Q3 and Q8) with female interviewees did not get a red light.

(h) Youth Employment Opportunities: All farms received a red light in this criterion, highlighting a need for providing opportunities for young people.

(i) Agrobiodiversity: Farms Q7 and Q8, with a higher number of crops, had a green light, while the others, with a red light, need to increase biodiversity.

(j) Soil Health: No farms received a red light, indicating good soil health. Farms Q1, Q6, and Q8, composed of a single plot, received a yellow light.

At last, with the application of ACT to the assessed farms, Farm Q8 attained the highest ACT score at 82%, whereas Farm Q4 achieved the lowest score at 27% (Table 6). The indicator with the highest score, registering at 94%, pertained to the 'Culture and Food Traditions' element. On the opposite end, the 'Responsible Governance' element, from Level 5: Rebuilding the Global Food System, scored 18% (Table 6).

**Table 6.** Results of ACT application on the eight farms (Q1 to Q8), in the center and south of Portugal, in 2022. Per-farm ACT provides the score (%) in each element of transition, linked with the levels of transition.

| Level of Transition | Element of Transition | Score (%) | | | | | | | | Medium Value per Indicator |
|---|---|---|---|---|---|---|---|---|---|---|
| | | Q1 | Q2 | Q3 | Q4 | Q5 | Q6 | Q7 | Q8 | |
| Level 1: Increase efficiency of industrial and conventional practices | 1.1. Efficiency | 43 | 43 | 43 | 29 | 57 | 29 | 57 | 86 | 48 |
| Level 2: Substitute industrial or conventional inputs with more sustainable alternatives | 2.1. Recycling | 17 | 17 | 33 | 17 | 33 | 0 | 33 | 83 | 29 |
| | 2.2. Regulation and balance | 20 | 20 | 50 | 20 | 20 | 10 | 50 | 100 | 36 |
| Level 3: Redesign whole agro-ecosystems | 3.1. Synergies | 25 | 25 | 38 | 25 | 50 | 0 | 38 | 88 | 36 |
| | 3.2. Diversity | 78 | 67 | 89 | 56 | 89 | 78 | 89 | 89 | 79 |
| | 3.3. Resilience | 33 | 33 | 33 | 33 | 33 | 67 | 67 | 100 | 50 |
| Level 4: Re-establish connections between growers and eaters; develop alternative food networks | 4.1. Circular and solidarity economy | 67 | 100 | 100 | 33 | 67 | 33 | 67 | 100 | 71 |
| | 4.2. Culture and food traditions | 100 | 100 | 100 | 50 | 100 | 100 | 100 | 100 | 94 |
| | 4.3. Co-creation and sharing of knowledge | 0 | 67 | 67 | 33 | 67 | 33 | 67 | 100 | 54 |
| Level 5: Rebuild the global food system so that it is sustainable and equitable for all | 5.1. Human and social value | 17 | 17 | 83 | 0 | 83 | 17 | 83 | 33 | 42 |
| | 5.2. Responsible governance | 0 | 40 | 20 | 0 | 20 | 20 | 20 | 20 | 18 |
| | Medium value per farm | 36 | 48 | 60 | 27 | 56 | 35 | 61 | 82 | |

Summarizing the key findings for each of the 11 elements of transition:

(a) Efficiency: Farm Q8 scored the highest, at 86%, while Farms Q4 and Q6 scored the lowest (29%). Farm Q8 has adopted practices such as water consumption reduction techniques, non-use of pesticides and veterinary drugs, efficient animal-feed utilization, solar energy adoption, and product dehydration and drying, as well as the use and conservation of traditional seeds. Farms Q4 and Q6 employed some of these efficient practices, though to a lesser extent.

(b) Recycling: Farm Q8 garnered the highest score, at 86%, whereas Farm Q6 attained the lowest score, at 0%. Farm Q8 uses alternative soil inputs, incorporates green manure, recycles wastewater, utilizes biomass residue for energy generation, and alternative methods of climate mitigation.

(c) Regulation and Balance (the element included in addition to the 10 from FAO): Farm Q8 achieved the highest score, at 100%, while Farm Q6 scored the lowest, at 10%. Farm Q8 implements a range of practices, including biological pest management, cover cropping for pest control and improved soil conditions, reduced tillage, adoption of organic and low-input farming, utilization of domesticated pollinators, and improved animal welfare and health. In contrast, Farm Q6 primarily relies on perennial crops.

(d) Synergies: Farm Q8 attained the highest score, at 88%, and Farm Q6 attained the lowest score, at 0%. Farm Q8's practices include non-crop plant cultivation, agroforestry, rotational/regenerative grazing, integrated crop–livestock systems, integrated pest management through habitat manipulation, and climate mitigation through system redesign.

(e)     Diversity: Farms Q3, Q5, Q7, and Q8 demonstrated the highest scores, at 89%, and Farm Q4 obtained the lowest score, at 56%. The high-scoring farms implement practices as improving local seed and breed diversity, integrating locally adapted crops and breeds, practicing polycultural cultivation, managing heterogeneous landscapes, and conserving the forest around agricultural lands. In contrast, Q4 places less emphasis on these practices.

(f)     Resilience: Five farms (Q1 to Q5) achieved the lowest score of 33%, by focusing primarily on livelihood resilience through diversified income sources. In contrast, Farm Q8 attained the highest score of 100%, due to its additional emphasis on systemic resilience to extreme weather events and disturbances, as well as its adaptive capacity to changing environmental conditions.

(g)     Circular and Solidarity Economy: Farms Q4 and Q6 received the lowest score of 33%, while Farms Q2, Q3, and Q8 achieved the highest score of 100%. This reflects their active involvement in supporting regional value generation, re-establishing the connection between producers and consumers, and their advocacy for seasonal and regional demand.

(h)     Culture and Food Traditions: All farms except Q4 achieved the highest score of 100%. These farms endorse healthy, diversified, and culturally appropriate food traditions and diets. Farm Q4, however, is not affiliated with farmers' associations or other platforms for policy support.

(i)     Co-Creation and Sharing of Knowledge: Farm Q8 scored the highest, at 100%, and Farm Q1 obtained the lowest score, at 0%. Farm Q8 actively engages with fellow farmers to exchange knowledge, participates in participatory and multi-stakeholder knowledge-generation approaches, and advocates for formal and informal production and food education.

## 4. Discussion

Revisiting the results, the indicators that scored the highest on the TAPE grid of indicators were 'Crops', from the 'Diversity' element, and 'Local or Traditional identity awareness', from the 'Culture and Food Tradition' element, with both achieving values of 3.0; meanwhile, the ones that scored the lowest were 'Renewable Energy' and 'Production' from the 'Recycling' element, with both obtaining values of 0.6. The criterion of the TAPE 10 Evaluation Criteria that received green lights from all farms was 'Land Tenure', and the one that received red lights from all farms was 'Youth employment opportunities'. Turning to ACT, the indicator that scored the highest was 'Culture and food traditions' (94%) and the one that scored the lowest was 'Responsible Governance' (18%).

Merging TAPE Grid of indicators, TAPE 10 Evaluation Criteria and ACT results are consistent for the best agroecological performance (Table 7). Farm Q8 scored the highest both in TAPE Grid of Indicators (2.8 values), had the highest number of green lights (six green lights), and scored the highest in ACT (82%), which means that its agroecological performance is better than that of the others. It is a regenerative family farm run by two young designers who transitioned from urban to rural life. They cultivate blueberries, vegetables, fruits, and eggs, distributing their produce through a CSA model. On the other hand, the farm that score the lowest on the TAPE grid of indicators was Farm Q6 (1.6 values), while the farm that had the highest number of red lights using TAPE 10 evaluation criteria was Farm Q5 (five red lights) and the farm that scored the lowest on the ATC was Farm Q4 (27%).

**Table 7.** Results of the TAPE grid of indicators' average values per farm, TAPE 10 evaluation criteria (number of green lights and number of red lights), and ATC medium value per farm, on eight farms (Q1 to Q8), in the center and south of Portugal, in 2022.

|  | Q1 | Q2 | Q3 | Q4 | Q5 | Q6 | Q7 | Q8 |
|---|---|---|---|---|---|---|---|---|
| Tape Grid of Indicators Average values per farm | 1.8 | 1.9 | 1.8 | 2.0 | 2.3 | 1.6 | 2.4 | 2.8 |
| Tape 10 Evaluation Criteria (number of green lights obtained) | 5 | 6 | 5 | 3 | 4 | 3 | 5 | 6 |
| Tape 10 Evaluation Criteria (number of red lights obtained) | 4 | 4 | 4 | 4 | 5 | 3 | 2 | 2 |
| ACT medium value per farm | 36 | 48 | 60 | 27 | 56 | 35 | 61 | 82 |

Upon comparing the values across the elements of transition in the TAPE grid of indicators (after converting to percentages) and in ACT, it becomes evident that while 'Culture and Food Transitions' received the highest score in both assessments (73% in the TAPE grid of indicators and 94% in ACT), the lowest scoring elements differ (Table 8). Specifically, 'Co-Creation and Sharing of Knowledge' achieved 23% in the TAPE grid of indicators (and 54% in ATC), while' Responsible Governance' scored 18% in ACT (and 54% in the TAPE grid of indicators), indicating a notable discrepancy.

**Table 8.** Results of the TAPE grid of indicators' average values, converted into percentages, and the ATC medium score by element of transition, on eight farms evaluated in the center and south of Portugal, in 2022.

| Elements/Elements of Transition | TAPE Grid of Indicators * | ACT |
|---|---|---|
| Diversity | 53 | 79 |
| Synergies | 53 | 36 |
| Efficiency | 53 | 48 |
| Recycling | 48 | 29 |
| Resilience | 55 | 50 |
| Culture and food traditions | 73 | 94 |
| Co-creation and sharing of knowledge | 23 | 54 |
| Human and social value | 50 | 42 |
| Circular and solidarity economy | 58 | 71 |
| Responsible governance | 48 | 18 |
| Regulation and balance | NA | 36 |

* values converted in %.

Nevertheless, several disparities are observed. For instance, when examining parameters related to young people and women, the focus on distinct aspects within the overarching theme leads to differential measurements. Regarding 'Youth', the TAPE grid of indicators' questions focus on youth empowerment and emigration, how young people see their future in agriculture, whether they are happy with the working conditions, or whether they intend to emigrate. The TAPE evaluation criteria add information about education and training, and ACT only mentions women and youth in the criterion 'Gender and Vulnerable Group Approach'. With regard to women, the TAPE indicator grid includes questions about women's empowerment, their access to decision-making power over productive resources, and the existence and functionality of women's organizations. The TAPE evaluation criteria add information on 'Control over the use of income'; 'Leadership in the community'; and 'Use of time'.

The analysis of the questions supporting the evaluation of the elements 'Culture and Food traditions', 'Co-creation and Sharing of Knowledge', and 'Responsible Governance', revealed that ACT incorporates more comprehensive inquiries. For instance, ACT includes questions about the right to adequate and culturally appropriate food, the support for individuals' decision-making regarding food sourcing and consumption (in 'Culture and Food traditions'), the farmers networks, the formal and non-formal education networks (in 'Co-creation and Sharing of Knowledge'), and the intersection of agroecology and global change (in 'Responsible Governance'), which are not included in the TAPE Grid of Indicators (Table 9).

**Table 9.** Indicators from the TAPE grid and transition criteria from the ACT for the elements Culture and Food Traditions, Co-creation and Sharing of Knowledge, and Responsible Governance.

| Elements | Tape Indicators | Act Transition Criteria |
|---|---|---|
| Culture and food traditions | Appropriate diet and nutrition awareness | Support healthy, diversified and culturally appropriate food traditions and diets |
| | Local or traditional identity and awareness | Support the right to adequate and culturally appropriate food |
| | Use of local varieties/breeds and traditional knowledge for food preparation | |
| Co-creation and sharing of knowledge | Platforms for the horizontal creation and transfer of knowledge and good practices | Connecting farmers to share knowledge |
| | Access to agroecological knowledge and interest of producers in agroecology | Promote formal and non-formal "production and food" education |
| | Participation of producers in networks and grassroot organizations | Promote participatory and multi-stakeholder approaches in knowledge generation |
| Responsible governance | Producers' empowerment | Policy development on producer-consumer links |
| | Producers' organizations and associations | Inclusive policy-making that aim for sustainable and equitable food system |
| | Participation of producers in governance of land and natural resources | Establishment of equitable governance and rights over natural resources |
| | | Policy development on the links between agroecology and global changes |
| | | Policy development that rewards agricultural management that enhances biodiversity and the provision of ecosystem services |

Contrary to the TAPE grid of indications (based on 35 indicators), the ACT is built around 62 transition criteria. The unique features of ACT are justified by this divergence. As opposed to ACT, which uses a binary "Yes or No" response format that could make it difficult to evaluate how agroecological performance on farms, the TAPE grid of indicators offers a more accurate assessment of indicator status—important information to evaluate the farms agroecological performance.

## 5. Conclusions

The TAPE grid of indicators, TAPE 10 evaluation criteria, and ACT collectively enable the assessment of agroecological performance of family farms. However, certain questions require adaptation to align with the Portuguese, and other European, cultural and territorial contexts. Examples include inquiries about women's associations, on-farm employability, local food systems, land inheritance, and emerging opportunities like agrotourism. The findings of this study are aligned with Anthonioz [63] who applied TAPE to livestock production in a region of France, pointing out that the results of some indicators, such as women's empowerment, may not be significant when applied to European agricultural systems. Marino [71], in their analysis of the food system in the city region of Rome (Italy),

applied an adaptation of the TAPE, noting its focus of application in developing countries. Steglich [72] in their work for the Global South, also applied an adaptation of TAPE in which they included indicators for assessing the presence of local non-agricultural sectors such as forestry, nature, and landscape conservation, renewable energies, and tourism In addition, Colbert [38] analyzed both TAPE and ACT within demonstration farms in Kenya and indicated that the high level of detail of TAPE, as well as the time technical knowledge required to study it, makes it less accessible to those not working in research. He claims that with slight modifications, ACT can be an easy tool for demonstration farms (as well as farmers) to assess how their activities are contributing to the agroecological transition, whether at the farm, field, market, or policy level.

In applying TAPE and ACT to Portuguese family farmers, questions pertaining to agricultural biodiversity and soil health pose challenges for farmers due to limited access to organized information. It is crucial to consider whether these constraints stem from factors related to the agroecosystem itself (such as size and cultural diversity), the local food system, or the characteristics of the farmers and stakeholders involved including their age, education, training, and time available. Both formal and informal education on agroecology practices (at the agroecosystem and food system level) are of the utmost importance yet, at present, are practically non-existent in Portugal. In the realm of family farming, the results indicate that the agroecological transition is primarily influenced by the socio-economic attributes of the farmers, rather than the geographic location or size/number of plots of the farms. Despite applying practices considered agroecological, such as polyculture and animal inclusion, most farmers lack familiarity with the concept of agroecology and possess a limited systemic understanding of agriculture and local food dynamics. Evaluating the agroecological performance of farms is just as important for farms already committed to the agroecological transition (consolidating existing efforts) as it is for those unfamiliar with the concept (supporting the transition process).

This study made it possible to identify the most relevant aspects for characterizing family farmers and their farms. It also made it possible to perceive the importance of how the evaluation criteria/indicators are ordered by element/theme, as it alters the values of each farm's agroecological performance. Thus, based on the analyses conducted and considering that the methodology to be developed aims to support the design of policies for agroecological transition, it was possible to identify a set of questions crucial for evaluating farms' agroecological performance, which might be organized into 10 topics (within the Agroecosystem and the Local/Global Food System):

Within the Agroecosystem:

(a)  Plant Production: evaluates the diversity of crops over space and time, considering factors like monoculture versus polyculture, as well as the integration of different crops. It also examines the origin of plants and seeds.
(b)  Animal Production: focuses on the presence and diversity of animals on the farm, emphasizing animal welfare. It considers factors such as multiple species, the integration of animals with crops, and their well-being.
(c)  Soil Regeneration: assesses practices related to recycling biomass, incorporating organic matter into the soil, soil cover, and conservation practices. It also considers the quantity of tillage activities.
(d)  Regenerating the Water Cycle: addresses water-related aspects, including the source of water, controlled irrigation, water recovery, and its destination. It also evaluates water quality.
(e)  Pest, Disease, and Weed Management: focuses on how the farm monitors and manages pests, diseases, and weeds. It considers decision-making processes and intervention methods.
(f)  Ecological Synergies (Functional Biodiversity): examines the integration of trees, crops, and animals. Additionally, it assesses the biodiversity on the farm as well as in its surroundings, including nearby farms.

(g) Economic Synergies (Management of Production Factors/Farm Management): considers the balance between internal and external factors in production. It looks at factors like seeds, plants, fertilizers, and farm machinery. It also includes record-keeping of various activities.

(h) Social Synergies (Well-Being on the Farm): evaluates aspects related to the well-being of individuals on the farm, including fair compensation, safety, rest time, gender equity, youth empowerment, and participation in decision-making processes. It also considers the farm's contribution to the community and its adherence to cultural and food traditions.

Within the Local/Global Food System:

(i) Interactions with the Local Food System: assesses the farm's relationships with the local community, including proximity to consumers, markets, and institutions. It also considers formal and informal networks between producers and producers and consumers.

(j) Sharing Local/Global Agroecological Knowledge: focuses on the sharing and co-construction of agroecological knowledge among various actors in the food system, both locally and globally.

The created methodology will involve conducting questionnaires with farmers during on-site visits, mapping out the farm areas, and performing soil analyses. The collected data will then be used to assess a set of indicators, combining elements from both the TAPE Grid of Indicators and ACT Criteria for Transition, positioning the indicators at the agroecosystem level or the food system. Additionally, discussions with respondents about the results will be held, a crucial step for farmers to take ownership of the findings. This process aids in identifying solutions and establishing collective paths forward, enhancing interaction among all involved stakeholders.

Two limitations of this study are important to mention. The first relates to the size of the sample: the questionnaires were applied to just eight farms. Despite the difficulties of applying them to a larger sample, especially due to the duration of the interviews, a larger number of assessed farms would have allowed for better understanding of the advantages and limitations of both TAPE and ACT methodologies in the Portuguese context. The second concerns the non-application of the four steps of the TAPE, which can condition the perception of the purpose of each question. Regarding TAPE Stage 0, due to the fact that the questionnaire was applied in two different regions, the gathering of local information relevant to the agroecological transition was not carried out. This information would be important for TAPE Stage 3, centered on returning to the territories to reflect on the results obtained, together with the farmers, and achieving a better understanding of agroecology. However, the conclusions reached for the creation of the methodology were in line with the results of other studies being carried out simultaneously in Portugal and with the bibliography of similar research being carried out in other countries. A more generalized application of these methodologies in Portugal aligned with other countries is, therefore, necessary in order to understand their adaptability in the context of family farming.

**Author Contributions:** Conceptualization: I.C.-P. and C.A.C. Investigation: I.C.-P., C.A.C., F.D. and A.A.R.M.A. All authors have read and agreed to the published version of the manuscript.

**Funding:** This research was funded by National Funds through the FCT—Foundation for Science and Technology, I.C.-P., within the scope of the individual doctoral research grant reference UI/BD/153087/2022.

**Institutional Review Board Statement:** The study was conducted in accordance with the Declaration of Helsinki, and approved by the Ethics Committee of Polytechnic of Viseu, approval code: Parecer no55/SUB/2021, date of approval: 22 June 2021.

**Informed Consent Statement:** Informed consent was obtained from all subjects involved in the study.

**Data Availability Statement:** Data are contained within the article.

**Acknowledgments:** We would like to thank CERNAS funded by national funds FCT—Foundation for Science and Technology, I.P., within the scope of the project Ref. UIDB/00681/2020, the GreenUPorto funded by national funds via FCT (Foundation for Science and Technology) through the Strategic Projects UIDB/05748/2020 and UIDP/05748/2020, DOI https://doi.org/10.54499/UIDP/05748/2020 and https://doi.org/10.54499/UIDB/05748/2020 and Valorizar a Agricultura Familiar Project (PDR2020-2024-058135) for the opportunity of applying the questionnaires within the scope of its research.

**Conflicts of Interest:** The authors declare no conflicts of interest.

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
