# Peer review of "A Methodological Framework for Assessing the Agroecological Performance of Farms in Portugal: Integrating TAPE and ACT Approaches"

_sustainability, doi:10.3390/su16103955_

Round 1
Reviewer 1 Report
Comments and Suggestions for Authors
The authors of the paper entitled "Methodological Framework to assess the Farms Agroecological Performance in Portugal: Integrating TAPE and ACT Approaches" have presented a detailed study for assessing the performance. However, I have the following comments.
Kindly advise authors to make necessary changes to their manuscript:
1) Discuss the limitations of the study:
Despite rigorous methodology and comprehensive analysis, every research work is subject to certain constraints that may impact the interpretation and generalizability of the findings. It is essential for authors to transparently acknowledge these limitations In the final section of the manuscript to ensure the integrity and validity of their work.
2) Describe the usefulness of the study for other geographical areas:
Create a new subsection on the rationale of the study under the introduction section and discuss the logic behind conducting this study along with its usefulness for other countries or regions such as rural places. This new subsection should help readers in understanding the motivation behind conducting this study as it would offer an insight into the context and justification for its implementation. In this section, authors should delve into elucidating the logic driving its conception and the potential benefits it offers to various stakeholders.
3) Considering the acronyms i.e. Tool for Agroecological Performance Evaluation (TAPE) and the Agroecological Criteria Tool (ACT) are integral to this paper's discussions. Hence, they should be referenced by their respective acronyms: TAPE and ACT once throughout the manuscript. In its present form, authors have repeatedly used full form along with its acronyms example: Page 6, line 200.
Author Response
Thank you very much for taking the time to review this manuscript. Please find the detailed responses below and the corresponding revisions in track changes and underlined in yellow in the re-submitted file.
Comments 1: Discuss the limitations of the study: Despite rigorous methodology and comprehensive analysis, every research work is subject to certain constraints that may impact the interpretation and generalizability of the findings. It is essential for authors to transparently acknowledge these limitations. In the final section of the manuscript to ensure the integrity and validity of their work.
Response 1: Thank you for pointing this out, we absolutely agree with this comment, thus we added in the conclusions the study limitations (lines 569-583)
Comment 2: Describe the usefulness of the study for other geographical areas: Create a new subsection on the rationale of the study under the introduction section and discuss the logic behind conducting this study along with its usefulness for other countries or regions such as rural places. This new subsection should help readers in understanding the motivation behind conducting this study as it would offer an insight into the context and justification for its implementation. In this section, authors should delve into elucidating the logic driving its conception and the potential benefits it offers to various stakeholders.
Response 2: Thank you again for this important insight. We have, accordingly, modified the introduction last paragraph to emphasize this point. (lines 91-101)
Comment 3: Considering the acronyms i.e. Tool for Agroecological Performance Evaluation (TAPE) and the Agroecological Criteria Tool (ACT) are integral to this paper's discussions. Hence, they should be referenced by their respective acronyms: TAPE and ACT once throughout the manuscript. In its present form, authors have repeatedly used full form along with its acronyms example: Page 6, line 200.
Response 3: Agree. We have changed using the acronyms
Reviewer 2 Report
Comments and Suggestions for Authors
The subject of this paper comparing 2 types of assessments of regenerative/ecological/sustainable agriculture is certainly worth merit. However, I feel it is not sufficiently clear to benefit the readers. There is no clear definition of the terms to start with such as agroecology/etc etc, and it seems that there is only 1 small farm/ horticultural enterprise. which qualifies ( Q8).
I think the paper should be shortened by at least 1/2 and rewritten with greater clarity.
I second their emphasis on women in agriculture, but they also need to emphasis more education ( since it seems most of their farmers were educated when farming was to maximize production at all costs, and so trained).
There is no history of the understanding of Organic Agriculture, what is it, and why, which is the basis of this further development ( eg Balfour Living Earth, Kiley-Worthington food first farming, ecological agriculture, 1993,Souvenir press, and others published in 50 to 80's).
The conclusions seem to infer that there should be an increase in women in agriculture, that social networking among farmers should expand, that incomes should increase and that biodiversity and reduced carbon footprint by increasing efficiency and net ( not gross) production which means changing the central belief of agriculture: "to maximize production at any cost"
Comments on the Quality of English LanguageSome curious English eg line 30: "it is stable, and increases annually (!)
Needs cutting and polishing
Author Response
Thank you very much for taking the time to review this manuscript. Please find the detailed responses below and the corresponding revisions in track changes and underlined in yellow in the re-submitted file.
Comments 1: The subject of this paper comparing 2 types of assessments of regenerative/ecological/sustainable agriculture is certainly worth merit. However, I feel it is not sufficiently clear to benefit the readers. There is no clear definition of the terms to start with such as agroecology/ etc etc, and it seems that there is only 1 small farm/ horticultural enterprise which qualifies (Q8).
Response 1: Thank you for pointing this out. We have defined agroecology in the introduction, but emphasizing on how to define an agroecological farm, as it is the focus of the study. Indeed, from the assessed farms, Q8 is the one with a better agroecological performance. The goal was to create a methodology to assess any farm, specially, family ones, with the purposes of helping farmers to improve their practices and results considering agroecological principles. This was also emphasized in the text.
Comments 2: I think the paper should be shortened by at least 1/2 and rewritten with greater clarity
Response 2: Thank you for the suggestion. As the study consists in the application and analyses of the two methodologies is not possible to be shortened by half. However, we tried to be more concise and clearer particularly in the Results and Discussion, reducing the length of the paper, without compromising the information provided.
Comments 3: I second their emphasis on women in agriculture, but they also need to emphasis more education (since it seems most of their farmers were educated when farming was to maximize production at all costs, and so trained).
Response 3: Thank you for this important insight. We have, accordingly, add the importance of education in the analysis (lines 246-252) and conclusion (lines 504-506)
Comments 4: There is no history of the understanding of Organic Agriculture, what is it, and why, which is the basis of this further development (eg Balfour Living Earth, Kiley-Worthington food first farming, ecological agriculture, 1993, Souvenir press, and others published in 50 to 80's).
Response 4: Thank you for pointing this out. We purposely didn't refer to typologies or production models such as organic farming, permaculture, biodynamic, regenerative or syntropic agriculture, because we considered that they all fit into the umbrella of agroecology and to discuss this subject would not fit on the scope of the present paper.
Comments 5: The conclusions seem to infer that there should be an increase in women in agriculture, that social networking among farmers should expand, that incomes should increase and that biodiversity and reduced carbon footprint by increasing efficiency and net (not gross) production which means changing the central belief of agriculture: "to maximize production at any cost."
Response 5: Thank you for pointing this out. The cost of maximising production without trying to re-establish natural resources has brought us to where we are. The paradigm must change from production to consumption, and agroecology can help in this process of transition to more sustainable food systems.
Comments 6: Some curious English eg line 30: "it is stable, and increases annually (!) "
Response 6: We have revised the sentence (Lines 260-261)
Point 1: Needs cutting and polishing.
Response to point 1: Thank you for the suggestion. We tried to be more concise and clearer reducing the length and polishing of the paper without compromising the information provided.
Reviewer 3 Report
Comments and Suggestions for Authors
What did you actually learn and conclude from this study? That needs to be made clear in the abstract and conclusion. The tables (esp. 1, 4 and 9) are too wordy and complex. The wording in line 129-130 needs to be clarified. Line 179: I was surprised to learn that only 8 farms were interviewed. Seems insufficient to evaluate so many methodologies. The entire article could/should be condensed? The scientific literature is well covered and used. I'm not sure how this analysis helps or improves the way to proceed in the future!?!
Author Response
Thank you very much for taking the time to review this manuscript. Please find the detailed responses below and the corresponding revisions in track changes and underlined in yellow in the re-submitted file.
Comments 1: What did you actually learn and conclude from this study? That needs to be made clear in the abstract and conclusion.
Response 1: Thank you for pointing this out, we absolutely agree with this comment, thus we added in the Abstract (lines 29-33) Conclusion the lessons learned with this manuscript (line 519-520)
Comment 2: The tables (esp. 1, 4 and 9) are too wordy and complex.
Response 2: Agree. We worked the tables, accordingly.
Comment 3: The wording in line 129-130 needs to be clarified.
Response 3: Agree. We have, accordingly add the missing information (lines 129-134)
Comment 4: Line 179: I was surprised to learn that only 8 farms were interviewed. Seems insufficient to evaluate so many methodologies.
Response 4: Thank you for pointing this out, we added in the Conclusions the study limitations (lines 569-583)
Comment 5: The entire article could/should be condensed.
Response 5: Thank you for the suggestion. We try to condense the article in the Results and Discussion without compromising the information provided
Comment 6: The scientific literature is well covered and used.
Response 6: Thank you.
Comment 7: I'm not sure how this analysis helps or improves the way to proceed in the future!?!
Response 7: Thank you again for this important insight. We have, accordingly, modified the introduction last paragraph to emphasize this point. (lines 91-100).
Round 2
Reviewer 1 Report
Comments and Suggestions for Authors
Accept in present form.
Author Response
Thank you very much for taking the time to review for the second time this manuscript.
Reviewer 2 Report
Comments and Suggestions for Authors
I have again read much of this paper and it is indeed improved somewhat. However it is still to lengthy although I take the point of the authors made to my comment.
Some examples from the conclusion
1451-142 already stated.
450-451 omit
511-513 rewrite this is NOT clear or concise. 518 omit "since it will....
573-576 Not clear, explain in one sentence or cut.
576 omit from "the last one....
Step 3. rewrite Reflect together with farmers to better understand agrocecology.
579 Conclusions for the methodology of evaluating farms agroecological performance in Portugal aligned with other countries, but a more general application to family farms is necessary.
Comments on the Quality of English Languagesame as above, it is now average importance, but still needs cutting rather than having a very wordy paper.
Author Response
Thank you very much for taking the time to review for the second time this manuscript. Please find the detailed responses below and the corresponding revisions in track changes and text underlined in blue in the re-submitted file.
Point-by-point response to Comments and Suggestions for Authors
Comments 1: I have again read much of this paper and it is indeed improved somewhat. However it is still to lengthy although I take the point of the authors made to my comment
Response 1: Thank you for the comment. We tried to be more concise and clearer reducing the length the paper without compromising the information provided.
Comment 2: 1451-142 already stated.
Response 2: We apologise but there must be a typo in the identification of the lines.
Comment 3: 450-451 omit
Response 3: Thank you for the suggestion. We have, accordingly omit that information.
Comment 4: 511-513 rewrite this is NOT clear or concise.
Response 4: Thank you for pointing this out, we have rewritten the sentence (underlined in blue).
Comment 5: 518 omit "since it will.....
Response 5: Thank you for the suggestion, we have rewritten the sentence (underlined in blue).
Comment 6: 573-576 Not clear, explain in one sentence or cut
Response 6: Thank you again for this important insight, we have rewritten the sentence (underlined in blue).
Comment 7: 576 omit from "the last one....
Response 7: Thank you for pointing this out, we have rewritten the sentence (underlined in blue) considering the comments 6 to 9.
Comment 8: Step 3. rewrite Reflect together with farmers to better understand agroecology.
Response 8: Thank you for pointing this out, we have rewritten the sentence (underlined in blue) considering the comments 6 to 9.
Comment 9: 579 Conclusions for the methodology of evaluating farms agroecological performance in Portugal aligned with other countries, but a more general application to family farms is necessary.
Response 9: Thank you again for this important insight, we have rewritten the sentence (underlined in blue) considering the comments 6 to 9.
Response to Comments on the Quality of English Language
Point 1: same as above, it is now average importance, but still needs cutting rather than having a very wordy paper.
Response to point 1: Thank you for the suggestion. We tried to improve the quality of the english language.